# MoReDrop: Dropout without Dropping

**Duo Li** [* 1]  **Li Jiang** [* 2]  **Yichuan Ding** [2]  **Xue Liu** [2]  **Victor Wai Kin Chan** [1]

## Abstract

Dropout is a widely adopted technique that significantly improves the generalization of deep neural networks in various domains. However, the discrepancy in model configurations between the training and evaluation phases introduces a significant challenge: the model distributional shift. In this study, we introduce an innovative approach termed Model Regularization for Dropout (MoReDrop). MoReDrop actively updates solely the dense model during training, targeting its loss function optimization and thus eliminating the primary source of distributional shift. To further leverage the benefits of dropout, we introduce a regularizer derived from the output divergence of the dense and its dropout models. Importantly, sub-models receive passive updates owing to their shared attributes with the dense model. To reduce computational demands, we introduce a streamlined variant of MoReDrop, referred to as MoReDropL, which utilizes dropout exclusively in the final layer. Our experiments, conducted on several benchmarks across multiple domains, consistently demonstrate the scalability, efficiency, and robustness of our proposed algorithms.

## 1. Introduction

In recent years, deep Neural Networks (DNNs) (Salakhutdinov, 2014; Schmidhuber, 2015) have made significant advancements across a wide range of areas such as computer vision, reinforcement learning, and natural language processing (Deng et al., 2009; Mnih et al., 2015; He et al., 2016; Vaswani et al., 2017; Ho et al., 2020; Jumper et al., 2021; Kang et al., 2024). While DNNs hold great promise with deeper networks (He et al., 2016; Wang et al., 2022), the

model complexity correspondingly escalates rapidly. This rapid escalation underscores the need for effective regularization techniques to mitigate overfitting and enhance the generalization capabilities of these deep models. Numerous strategies have been developed to tackle these challenges, with dropout gaining prominence due to its simplicity and efficacy extensively utilized in many recent AI breakthroughs (Hinton et al., 2012; Srivastava et al., 2014; Dosovitskiy et al., 2021; Jumper et al., 2021; Ramesh et al., 2022; Wang et al., 2024).

Dropout generally uses a Bernoulli-distributed mask applied to each layer before each training step, which also implies independently and randomly deactivating each neuron with probability $p$ but activating all neurons during inference. Intuitively, dropout training mimics, simultaneously and jointly, training an ensemble of neural networks with varying deactivated unit configurations, all while utilizing shared weights and parameters to the dense model (Hinton et al., 2012; Srivastava et al., 2014). However, it is non-trivial to explicitly assemble sub-models[1], and a single model characterized by scaled parameters without dropout, i.e., the dense model, is employed for the practical evaluation period. It introduces a subtle but non-negligible model distributional shift between training and evaluation stages.

A variety of regularizers have been proposed to mitigate this issue. Sub-to-sub regularization paradigm imposes constraints between pairs of sub-models in the training phase. The primary objective of this constraint is to ensure consistency across different sub-models, thereby maintaining the expectation of a unified, coherent model for evaluation. Examples of such regularizers include $L_2$ distance (Zolna et al., 2018), Kullback-Leibler (KL) divergence for two random sub-models (Liang et al., 2021) or worse-case sub-models (Xia et al., 2023). Another line of research introduces the dense-to-sub regularization (Ma et al., 2017). This approach imposes constraints from the divergence of dense and sub-models throughout the training process, ensuring consistency across the pair of dense-sub models. However, existing methodologies consistently employ an active update strategy for sub-models throughout the training process, with a predominant focus on minimizing sub-models loss.

---

[*]Equal contribution  [1]Tsinghua University, China  [2]McGill Unversity, China. Correspondence to: Duo Li <ld21@mails.tsinghua.edu.cn>, Li Jiang <li.jiang3@mail.mcgill.ca>.

*2nd Workshop on Advancing Neural Network Training: Computational Efficiency, Scalability, and Resource Optimization, Proceedings of the $41^{st}$ International Conference on Machine Learning*, Vienna, Austria. PMLR 235, 2024.

---

[1]We note sub-models as their corresponding dense model employed dropout with shared parameters.

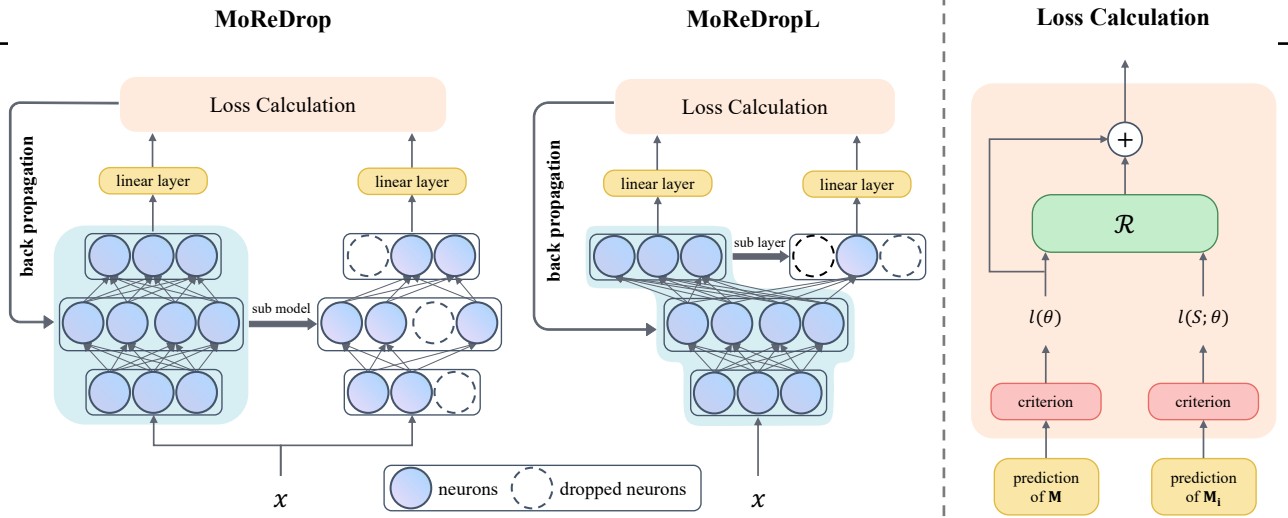

*Figure 1.* The overall framework and detailed loss computation of MoReDrop. **Left:** The input data $x$ is processed twice, once each through the dense model and its sampled sub-model, yielding two outputs. The loss function of the dense model is then regularized by minimizing the discrepancy between these two outputs. The gradient backpropagation is performed only on the dense model ( blue-shaded area ). **Right:** The details for calculating the regularized loss for the dense model.

Those approaches fail to eliminate the model distributional shift, which is an intrinsic consequence of the expectation operation employed during evaluation.

In this study, we propose a novel dense-to-sub regularization approach, named Model Regularization for Dropout (MoRe-Drop), to entirely mitigate the model distributional shift. MoReDrop *exclusively* updates the dense model, thereby circumventing the expectation operator to guarantee a uniform model configuration across both training and inference stages, with a primary emphasis on its loss function. Moreover, MoReDrop introduces a novel regularization term, endowed with a bounded property, into the loss function. This term is designed to quantify the discrepancy in outputs between the dense model and a randomly sampled sub-model for the same mini-batch dataset.

Remarkably, the regularizer functions as a toolkit, enabling the dense model to exploit the advantages of dropout at each gradient iteration, but without explicit dropout implementation. Conversely, the sub-models receive passive updates via parameter sharing with the dense model over the course of training. The overall framework and training details of MoReDrop are shown in Fig. 1.

However, same as other methods, MoReDrop requires an additional forward pass in the sub-model, which is computationally intensive for parameter-heavy deep layers (Devlin et al., 2019; Dosovitskiy et al., 2021). To alleviate this, MoReDropL is introduced, reducing the computational demand to a single last-layer forward pass by applying dropout solely to the final shared-parameter layer. Intuitively, MoRe-DropL trades a marginal loss in generalization for significantly lower computational costs.

We assess our proposed methods across various models and tasks, containing image classification, image generation tasks, and language understanding. Experiments show that our proposed methods allow for a higher dropout rate, which potentially further improves the performance but avoids the distributional shift. We also observe that MoReDrop consistently delivers superior performance compared to state-of-the-art baselines. Surprisingly, MoReDropL also surpasses previous methods in many tasks though it trades off the model generalization ability for computation efficiency.

## 2. Related Work

**Dropout and its Variants.** Regularization plays a pivotal role in preventing overfitting in deep learning and large-scale models. A multitude of regularization techniques has been proposed to address this issue, including but not limited to weight decay, dropout, batch normalization, noise addition, early stopping, and label smoothing (Simonyan & Zisserman, 2015; Ioffe & Szegedy, 2015; Poole et al., 2014; Yao et al., 2007; Szegedy et al., 2016). Among these, dropout stands out as a particularly effective method due to its simplicity and broad applicability in different domains (Hinton et al., 2012; Srivastava et al., 2014). For different model architectures, different dropout methods a variety of dropout methods have been proposed. These consist of DropConnect (Wan et al., 2013) for fully connected layers, SpatialDropout (Tompson et al., 2015) and DropBlock (Ghiasi et al., 2018) for convolutional neural Networks, DropPath (Larsson et al., 2017) for ResNet, and DropHead (Zhou et al., 2020) for Transformer models. In addition to its role as a regularization method to prevent overfitting, dropout has also been utilized as a data augmentation technique (DeVries & Taylor, 2017; Zhong et al., 2020), further contributing to its effectiveness and versatility.

*Table 1.* Comparison of different regularization approaches to mitigate the model distributional shift between the training and inference stages. It is important to highlight that the distributional gap of MoReDrop between the models in the inference and training stages is zero because the same model configuration, i.e., the dense model, has been used for those stages.

| Algorithms | Dense model update | Sub-model update | Regularizer | Distance function | Distribution Shift |
|---|---|---|---|---|---|
| FD (Zolna et al., 2018) | Passive | Active | Sub-to-sub | $L_2$ | $\mathcal{G}$ |
| R-Drop (Liang et al., 2021) | Passive | Active | Sub-to-sub | KL | $\mathcal{G}$ |
| WordReg (Xia et al., 2023) | Passive | Active | Sub-to-sub | KL | $\mathcal{G}$ |
| ELD (Ma et al., 2017) | Passive | Active | Dense-to-sub | $L_2$ | $\mathcal{G}$ |
| MoReDrop (ours) | Active | Passive | Dense-to-sub | Modified $L_1$ (Eqn. (4)) | 0 |

**Model Distributional Shift.** Prior work has revealed that dropout brings the inconsistency between training and inference stages, specifically, the model distributional shift. There are primary two categories to address this issue: (1) Sub-to-sub regularization paradigm aims to maintain the consistency between a pair of sub-models in the training process. In this paradigm, Fraternal Dropout (FD) (Zolna et al., 2018) employs $L_2$ distance on hidden states, R-Drop (Liang et al., 2021) utilizes the on two sampled sub-models with dropout, and Worst-Case Drop Regularization (WordReg) (Xia et al., 2023) holds the same inspiration with R-Drop but firstly find the two worse-case sub-models. (2) dense-to-sub regularization paradigm, i.e., Expectation Linear Dropout (ELD) (Ma et al., 2017), which maintains the consistency between a pair of dense-sub models as training progresses. MoReDrop belongs to the dense-to-sub regularization paradigm while still holding a significant difference.

**Knowledge Distillation.** Our research delves into the regularization of outputs between two distinct models: a dense model and a sub-model employing dropout. This approach resonates with the principles of knowledge distillation and the teacher-student training paradigm (Hinton et al., 2015; Romero et al., 2014; Yim et al., 2017; Allen-Zhu & Li, 2020; Chen et al., 2020; Tarvainen & Valpola, 2017). Contrary to prior research in knowledge distillation, our approach does not explicitly define which model serves as the teacher or the student. In our framework, the dense model is expected to provide a stable baseline performance, with further improvements realized through regularization between the dense and sub-models. This dual functionality allows either model to assume the role of teacher or student. However, our primary objective is to enhance the generality and robustness of the dense model, positioning it as the student in this context. Notably, our methodology does not introduce additional parameters or a pre-trained teacher model during the training process, positioning it as a form of self-distillation (Xu & Liu, 2019; Zhang et al., 2019; Mobahi et al., 2020; Allen-Zhu & Li, 2020).

## 3. Preliminaries

**Notation** The training set, denoted as $\mathcal{D}$, consists of pairs $\{(x_1, y_1), \ldots, (x_N, y_N)\}$, where $N$ signifies the total number of pairs in $\mathcal{D}$. In this context, each pair $(x_i, y_i)$ in $\mathcal{D}$ is typically considered an independent and identically distributed (i.i.d.) sample drawn from the respective distributions of $X \in \mathcal{X}$ and $Y \in \mathcal{Y}$, respectively.

Consider a DNN, denoted by $\mathbf{M}$, consisting of $L$ hidden layers, with $X$ and $Y$ representing the input and output, respectively. Each layer in the network is indexed by $l$, which spans from 1 to $L$. The output vector from the $l^{th}$ layer is signified by $\mathbf{h}^{(l)}$. In this setup, the network's input is specified as $\mathbf{h}^{(0)} = x$, and the final network output is $\mathbf{h}^{(L)}$. The network $\mathbf{M}$ is characterized by a set of parameters collectively symbolized by $\theta = \{\theta_l : l = 1, \ldots, L\}$. Here, $\theta_l$ encapsulates the parameters associated with the $l^{th}$ layer. With slight abuse of notation, we indicate $l(\theta)$ as the loss function.

**Dropout** In the naïve dropout formulation (Hinton et al., 2012; Srivastava et al., 2014), each layer is associated with $\Gamma^{(l)}$, a vector composed of independent Bernoulli random variables. Each of these variables has a probability $p_l$ of taking the value 0 and a probability $1 - p_l$ of assuming the value 1. This is analogous to independently deactivating the corresponding neuron (effectively setting each weight to zero) with a probability $p_l$. We introduce a set of dropout random variables, denoted by $S = \{\Gamma^{(l)} : l = 1, \ldots, L\}$, where $\Gamma^{(l)}$ corresponds to the dropout random variable for the $l^{th}$ layer. We can represent the deep neural network $\mathbf{M_i}$ as:

$$\mathbf{h}^{(l)} = f_l(\mathbf{h}^{(l-1)} \odot \gamma^{(l)}),$$

where $\odot$ denotes the element-wise product, and $f_l$ represents the transformation function for the $l^{th}$ layer. For instance, if the $l^{th}$ layer is a fully connected layer with a weight matrix $W$, a bias vector $b$, and a sigmoid activation function $\sigma(x) = 1/(1 + \exp(-x))$, then the transformation function is defined as $f_l(x) = \sigma(Wx + b)$. To justify the connection between the dense model and the sub-model, we represent $\mathbf{M_i}$ as the sub-model derived from $\mathbf{M}$ through the application of dropout, $i$ can be any number to represent different sub-models. We also use $\mathbf{h}^{(l)}(x, s; \theta)$ to denote

the output of the $l^{th}$ layer given the input $x$ and the dropout value $s$, under the parameter set $\theta$.

Conceptually, dropout seeks to train an ensemble of exponentially many neural networks concurrently, with each network corresponding to a unique configuration of deactivated units, while sharing the same weights or parameters (Hinton et al., 2012; Hara et al., 2016). We denote the loss function with dropout as $l(S; \theta)$.

**Model distributional shift.** Dropout implicitly forms an ensemble of neural networks via weight sharing during the training phase. However, for evaluation, a deterministic dense model without dropped neurons is employed to approximate the ensemble operation. This approximation results in a model distributional shift, noted as $\mathcal{G}$, between training and evaluation when dropout is used:

$$\mathcal{G} = \mathbb{E}_S \left[ \mathbf{H}^{(L)}(x, S; \theta) \right] - \mathbf{h}^{(L)}(x, \mathbb{E}[S]; \theta), \quad (1)$$

where the LHS of the minus signifies the ideal ensemble model with dropout for evaluation, which is represented by a dense model with expected activate units on the RHS of the minus.

# 4. Model Regularization for Dropout

In this section, we delve into the specifics of our proposed algorithm which distinctly focuses on actively updating the dense model with dense-to-sub regularization, MoReDrop and MoReDropL. The overall high-level structure is depicted in Fig. 1. We then proceed to discuss the chosen regularizer with its analysis and practical implementation of MoReDrop. Finally, we draw a comparison between our algorithm and the prior methods to alleviate the model distributional shift, elucidating why our proposed algorithm exhibits superior performance, even under high dropout rates.

## 4.1. Actively Updating the Dense Model via Its Loss

In supervised learning, for standard dense model training, the learning objective is to minimize the following negative log-likelihood function:

$$\frac{1}{N} \sum_{i=1}^{N} l(\theta) = -\frac{1}{N} \sum_{i=1}^{N} \log p\left(y_i \mid x_i; \theta\right). \quad (2)$$

For dropout training, the optimization object additionally incorporates the marginalization of the dropout variables (Wang & Manning, 2013; Srivastava et al., 2014; Ma et al., 2017):

$$\mathbb{E}_{S_i} \left[ \frac{1}{N} \sum_{i=1}^{N} l(S; \theta) \right] = -\mathbb{E}_{S_i} \left[ \frac{1}{N} \sum_{i=1}^{N} \log p(y_i | x_i, S_i; \theta) \right]. \quad (3)$$

The pivotal element driving the distributional shift is the disparity between model configurations during training and inference. During training, there is an *active* updating of sub-models, while inference is conducted with a deterministic model. This approach implicitly defines an expectation term and consequently introduces a model distributional shift (Ma et al., 2017; Liang et al., 2021), as delineated in Eqn. (1). To mitigate this discrepancy, we introduce a novel method, termed Model Regularization for Dropout (MoReDrop). Our approach begins by conducting gradient backpropagation exclusively on the dense model, prioritizing the loss from this configuration without dropout, shown in Eqn. (2). Unlike conventional methods that pursue model consistency solely during training, MoReDrop ensures model consistency applied throughout both the training and inference stages. This approach aligns with traditional dense training, thus nullifying the model distributional shift: $\mathcal{G} = 0$.

## 4.2. Dense-to-Sub Regularization

However, the advantages of dropout have been forsaken. To incorporate these benefits, our algorithm imposes constraints on the pair consisting of the dense model $\mathbf{M}$ and a randomly sampled sub-model $\mathbf{M_i}$. This is achieved through *passive* updating—without gradient backpropagation by exploiting their shared-parameter relationship with the dense model.

$$\mathcal{R} = g(l(S; \theta) - l(\theta)),$$

Typically, both the KL divergence and $L_2$ divergence are unbounded and may outweigh the primary loss function from the dense model, particularly in scenarios characterized by high dropout ratios. Moreover, these divergences exhibit heightened sensitivity to the coefficient settings.

In the present study, we adopt the function $g = (\exp(\alpha \cdot x) - 1)/(\exp(\alpha \cdot x) + 1)$, a variant of the Logistic Sigmoid function to confine the output within the range $(-1, 1)$ and $g(0) = 0$, where $\alpha$ serves as an exponential-inner coefficient for $\mathcal{R}$. This specific formulation of the regularization term offers two notable advantages over other divergences or distance functions. Firstly, its bounded nature imparts robustness to the loss function against varying dropout rates $p$, rendering it possible to train a near-optimal model even at high values of $p$. Second, the incorporation of the coefficient $\alpha$ into the exponential function standardizes the hyperparameter search space across various tasks, with the dense model providing a stable "baseline" performance, upon which further enhancements are realized through this regularization. Specifically, the regularization term we employed is as follows:

$$\mathcal{R} = \frac{\exp(\alpha \cdot (l(S; \theta) - l(\theta))) - 1}{\exp(\alpha \cdot (l(S; \theta) - l(\theta))) + 1}, \quad (4)$$

and the final optimization objective in our algorithm is:

$$\underset{\theta}{\arg\min} -\frac{1}{N}\sum_{i=1}^{N}\left(\log p\left(y_i \mid x_i; \theta\right) - \mathcal{R}\right), \qquad (5)$$

where it is bifurcated into two principal elements: the primary one being the loss of the dense model, which guarantees model consistency for both training and inference; and a supplementary regularization term $\mathcal{R}$, incorporated to draw upon the advantages offered by dropout, particularly generalization abilities. Importantly, MoReDrop conducts gradient backpropagation exclusively on the dense model, yet it still capitalizes on the advantages of dropout models.

An ablation study on various regularization forms, as detailed in Appendix B.7, showcases the superiority of our designed $\mathcal{R}$. Nevertheless, we recognize the potential for more effective loss functions and consider their exploration an avenue for future research.

### 4.3. Regularizer Analysis

As the concrete form of $\mathcal{R}$ in Eqn. (4) cannot guarantee a point-wise lower bound to 0, i.e., $\exists x, y$, s.t. $g(l(S; \theta) - l(\theta)) \leq 0$, which results in $l(S; \theta) - l(\theta) \leq 0$. In theory, this could potentially lead to hacking solutions, such as an infinitely large $l(\theta)$ alongside a near-zero $l(S; \theta)$, against the primary optimization objective to minimize $l(\theta)$. However, that scenario is precluded because (1) the upper bound nature of $g(\cdot)$ and the coefficient $\alpha$ limit the magnitude of $\mathcal{R}$; and (2) due to the passive update process of sub-models from the dense model, the value of $l(S; \theta)$ tends to stay proximate to $l(\theta)$ within a reasonable range.

In our experiments, we find that the expectation of $l(S; \theta) - l(\theta) \geq 0$ holds across diverse dropout rate configurations, as shown in Fig. 7. In contrast, the expected value of the concrete form of $\mathcal{R}$ over the set $S$, taken across the entire training dataset, is provably non-negative when $l(S; \theta) - l(\theta) \leq 0$, as established in Theorem 4.1 (see details in Appendix A), which further precludes undesirable solutions, i.e., $l(\theta) \to \infty$.

**Theorem 4.1.** *The regularizer $\mathcal{R}$ (Eqn. (4)), under the condition of $l(S; \theta) - l(\theta) \leq 0$, maintains non-negativity over the expectation over $S$ throughout the training process:*

$$\mathbb{E}_{S_i}\left[\frac{1}{N}\sum_{i=1}^{N}\frac{\exp(\alpha \cdot (l(S; \theta) - l(\theta))) - 1}{\exp(\alpha \cdot (l(S; \theta) - l(\theta))) + 1}\right] \geq 0$$

### 4.4. Algorithm Summary

During every gradient update, we execute the forward operation twice using the same randomly sampled mini-batch dataset: once for the dense model and once for the sub-model. Then, we calculate the loss function based on these two forward operations. We summarize our final algorithm in Algorithm 1 and Fig. 1. Note that we only apply gradient update to the dense model $\mathbf{M}$; the sub-model $\mathbf{M_i}$ does not undergo updates through gradient backpropagation and we note the stop gradient operator as $\square[\cdot]$. To further mitigate

---

**Algorithm 1** Model Regularization for Dropout

**Input:** $\mathcal{D} = \{(x_i, y_i)\}_{i=1}^{N}$, coefficient $\alpha$.
Initialize $\mathbf{M}_\theta$
**for** $t = 1$ **to** $N$ **do**
    Randomly sample mini-batch $B_i \sim \mathcal{D}$
    Randomly sample a sub-model $\mathbf{M_{i\theta}}$
    Forward the $B_i$ to the dense model $\mathbf{M}_\theta$ and obtain $l(\theta)$
    Forward the $B_i$ to the sub-model $\mathbf{M_{i\theta}}$ and obtain $\square[[l(S; \theta)]$
    Update $\mathbf{M}_\theta$ based on Eqn. (5)
**end for**

---

computational costs, we introduce a light variant of MoReDrop, termed MoReDropL. This version retains the guiding principle of MoReDrop, where the main loss comes from the dense model regularized by the interplay between the dense model and its sub-models. The key distinction between MoReDropL and MoReDrop lies in their network structure for utilizing dropout: MoReDropL employs dropout solely in the *final* layer, thereby circumventing additional matrix computations in forward, while MoReDrop applies dropout across *all* layers, necessitating extra matrix computations for all networks. Although MoReDropL sacrifices a degree of generalization capability, this compromise enables a significant reduction in computational burden.

### 4.5. Discussions

Our proposed method, MoReDrop and MoReDropL, exhibits similarities with various established approaches handling model distributional shift raised from the employment of dropout, as detailed in Table 1. However, MoReDrop finds a distinct way to mitigate the model distributional shift and follows two principle ordering pipelines: a). Firstly, actively updating solely on the dense model during training to zero-forcing the model distributional shift b). Further, introducing the regularizer to embrace the benefits of dropout.

**Mitigation of Model Distributional Shift.** In MoReDrop, model configurations during training and inference are identical to ensure $\mathcal{G} = 0$ and passively update sub-models. In contrast, alternative methods that actively update sub-models incur an implicit expectation and thus $\mathcal{G} > 0$, despite various regularization attempts to minimize it. Furthermore, the dense model actively updating imparts a notable resilience to variations in dropout ratios, which is inherently independent of both $p$ and $\alpha$, thereby ensuring stable base performance under different coefficient configurations.

*Table 2.* The averaged results of NLU tasks on the GLUE benchmark. MoReDrop outperforms the backbone model for all tasks and MoReDropL outperforms the backbone in 17 out of 24 tasks. The best performances are in **bold**.

| Methods | CoLA Matt. | SST-2 Acc. | MRPC Acc./F1 | STS-B P. Corr. | QQP Acc./F1 | MNLI m./mm. | QNLI Acc. | RTE Acc. | Average |
|---|---|---|---|---|---|---|---|---|---|
| BERT-base | $56.49_{\pm0.24}$ | $93.31_{\pm0.12}$ | $85.10_{\pm0.31}$ / $89.41_{\pm0.18}$ | $87.92_{\pm0.22}$ | $91.38_{\pm0.02}$ / $87.55_{\pm0.02}$ | $83.49_{\pm0.16}$ / $84.84_{\pm0.18}$ | $91.46_{\pm0.12}$ | $67.99_{\pm0.21}$ | 83.54 |
| + MoReDropL | $58.23_{\pm0.39}$ | $92.52_{\pm0.07}$ | $87.12_{\pm0.22}$ / $90.82_{\pm0.29}$ | $88.24_{\pm0.18}$ | $91.21_{\pm0.04}$ / $87.77_{\pm0.09}$ | $83.97_{\pm0.11}$ / $84.46_{\pm0.11}$ | $91.14_{\pm0.12}$ | $69.05_{\pm0.13}$ | 84.05 |
| + MoReDrop | $\mathbf{58.99}_{\pm0.26}$ | $\mathbf{93.53}_{\pm0.10}$ | $\mathbf{87.18}_{\pm0.31}$ / $\mathbf{90.86}_{\pm0.22}$ | $\mathbf{88.31}_{\pm0.09}$ | $\mathbf{91.41}_{\pm0.04}$ / $\mathbf{87.97}_{\pm0.04}$ | $\mathbf{84.98}_{\pm0.21}$ / $\mathbf{85.27}_{\pm0.22}$ | $\mathbf{91.59}_{\pm0.14}$ | $\mathbf{69.98}_{\pm0.27}$ | **84.55** |
| RoBERTa-base | $60.07_{\pm0.22}$ | $93.86_{\pm0.28}$ | $87.50_{\pm0.33}$ / $90.84_{\pm0.21}$ | $89.68_{\pm0.22}$ | $91.02_{\pm0.07}$ / $87.40_{\pm0.11}$ | $87.77_{\pm0.11}$ / $87.49_{\pm0.24}$ | $92.62_{\pm0.07}$ | $72.77_{\pm0.33}$ | 85.55 |
| + MoReDropL | $\mathbf{62.39}_{\pm0.31}$ | $94.09_{\pm0.41}$ | $88.19_{\pm0.54}$ / $91.52_{\pm0.33}$ | $90.46_{\pm0.27}$ | $91.38_{\pm0.05}$ / $87.94_{\pm0.18}$ | $87.20_{\pm0.13}$ / $86.71_{\pm0.27}$ | $92.27_{\pm0.02}$ | $\mathbf{79.48}_{\pm1.33}$ | 86.51 |
| + MoReDrop | $62.37_{\pm0.33}$ | $\mathbf{94.79}_{\pm0.37}$ | $\mathbf{89.80}_{\pm0.22}$ / $\mathbf{92.44}_{\pm0.16}$ | $\mathbf{90.55}_{\pm0.16}$ | $\mathbf{91.55}_{\pm0.09}$ / $\mathbf{88.17}_{\pm0.09}$ | $\mathbf{87.90}_{\pm0.10}$ / $\mathbf{87.60}_{\pm0.17}$ | $\mathbf{92.73}_{\pm0.11}$ | $77.45_{\pm0.41}$ | **86.85** |
| DeBERTaV3-xsmall | $61.34_{\pm0.82}$ | $92.60_{\pm0.37}$ | $89.58_{\pm0.92}$ / $92.29_{\pm0.78}$ | $\mathbf{89.95}_{\pm0.16}$ | $90.54_{\pm0.04}$ / $86.73_{\pm0.04}$ | $87.84_{\pm0.08}$ / $87.94_{\pm0.12}$ | $\mathbf{92.47}_{\pm0.16}$ | $71.02_{\pm0.50}$ | 85.66 |
| + MoReDropL | $63.46_{\pm1.12}$ | $\mathbf{93.12}_{\pm0.35}$ | $89.64_{\pm0.14}$ / $92.31_{\pm0.15}$ | $\mathbf{89.95}_{\pm0.24}$ | $90.50_{\pm0.11}$ / $86.65_{\pm0.08}$ | $87.91_{\pm0.04}$ / $87.64_{\pm0.02}$ | $92.37_{\pm0.22}$ | $71.18_{\pm0.51}$ | 85.88 |
| + MoReDrop | $\mathbf{64.64}_{\pm1.68}$ | $92.93_{\pm0.11}$ | $\mathbf{90.04}_{\pm0.36}$ / $\mathbf{92.74}_{\pm0.23}$ | $90.32_{\pm0.02}$ | $\mathbf{91.21}_{\pm0.04}$ / $\mathbf{87.70}_{\pm0.10}$ | $\mathbf{88.07}_{\pm0.08}$ / $\mathbf{88.14}_{\pm0.09}$ | $92.49_{\pm0.11}$ | $\mathbf{76.46}_{\pm1.06}$ | **86.79** |

**Motivation for the regularizer.** With the same formulation from the mathematical side to minimize the output discrepancy, the regularizer of other approaches aligns with the learning objective to reduce the model distributional gap, either dense-to-sub or sub-to-sub. Conversely, the purpose of $\mathcal{R}$ in MoReDrop is to leverage the advantages of dropout.

**Limitation of sub-to-sub regularization.** Sub-to-sub regularization, such as R-Drop, potentially struggles with the polynomial growth of model pairs, particularly under high dropout rates. Thus, sub-to-sub regularization methods may inadvertently compromise model generalization in favor of maintaining sub-model consistency due to their larger search spaces. To confirm this, a systematic analysis through the training loss curves is presented in Section 5.4.

## 5. Experiments

To underscore the wide-ranging applicability of our proposed method, we conducted a thorough evaluation spanning distinct machine learning domains, i.e., general language understanding, image classification, and image generation with different backbones algorithms (Section 5.1, Section 5.2, and Section 5.3). In the Section 5.4, we present a systematic comparison of our dense-to-sub regularization with R-Drop sub-to-sub regularization. This comparative study provides a crucial understanding of the superior performance of MoReDrop. Further experimental details, training time, and ablations on hyperparameters and forms of regularization are provided in Appendix B.

To ensure a fair comparison, we retained the same common training hyperparameter setting as in the baseline models, such as epochs and batch size. We prioritize R-Drop as the baseline due to its efficiency and superiority over others, following the same setting as R-Drop. The absence of certain baseline results is a deliberate choice with different reasons, detailed in Appendix B.8.

### 5.1. Natural Language Understanding

**Benchmark Datasets.** Our evaluation begins with natural language understanding tasks, for which we apply our proposed methods on the standard development sets of the General Language Understanding Evaluation (GLUE) benchmark (Wang et al., 2019). The GLUE benchmark includes eight unique tasks, all of which involve text classification or regression. The distinct characteristics of each task provide a comprehensive and robust testing ground for our proposed methodology. The evaluation metrics for the eight tasks and experiment details are shown in Appendix B.2.

**Model & Training.** We utilize three publicly available pre-trained models: BERT-base (Devlin et al., 2019) and RoBERTa-base (Liu et al., 2019) and DeBERTaV3-xsmall (He et al., 2020; 2021) as our foundation models for fine-tuning. As BERT-base, RoBERTa-base and DeBERTaV3-xsmall all employ standard dropout, we could directly apply our method and use the original models for comparison. For MoReDrop, we performed a parameter sweep, setting the dropout probability $p$ to span from 0.1 to 0.9 and $\alpha$ to take on values from the set $\{0.1, 0.5, 1, 2\}$, uniformly applying this sweeping across all subsequent domains.

**Results.** We present the final performance in Table 2 averaged by 5 independent seeds. MoReDrop outperforms the baselines across all tasks, showing averaged improvements of approximately 1.0%, 1.3% and 1.1% on BERT-base, RoBERTa-base and DeBERTaV3-xsmall models, respectively. Our findings also indicate that MoReDropL enhances average performance over the baseline. Interestingly, MoReDropL outperforms MoReDrop on some tasks, notably achieving a 2% performance margin on the RTE task. We hypothesize that applying dropout to all layers may disrupt beneficial features in pre-trained layers, while limiting dropout to the last layer preserves these features, enabling more effective task-specific fine-tuning. This will be investigated in future work.

## 5.2. Image Classification Domains

**Benchmark Datasets.** Our image classification experiments were conducted on three well-recognized benchmark datasets: CIFAR-10, CIFAR-100 (Krizhevsky et al., 2009) and ImageNet (Deng et al., 2009). The CIFAR-10 and CIFAR-100 datasets both consist of low-dimensional pixel images, with the primary distinction between them being the number of categories they feature, as indicated by their respective names. Conversely, the ImageNet dataset presents a significantly greater challenge, encompassing more than $1,000$ categories.

**Model & Training.** To offer a scalable comparison of MoReDrop in the domain of image classification, we utilize three models: a small model with 1.2 million parameters (ResNet-18) (He et al., 2016), a medium model with 86 million parameters (ViT-B/16) (Dosovitskiy et al., 2021) and a large model with 307 million parameters (ViT-L/16). Note that the standard ResNet-18 does not incorporate dropout, while the default configuration of the vanïlla ViT-B/16 and ViT-L/16 set $p = 0.1$. For the ResNet-18, our baselines comprise: (1) DropBlock (Ghiasi et al., 2018), which mitigates overfitting by dropping continuous regions of neurons, and (2) DropPath (Larsson et al., 2017), which zeroes out an entire branch in the neural network during training, aiming to achieve the same goal as DropBlock. It is worth noting that ResNet incorporates batch normalization as a technique to combat overfitting, disrupting the training process (Ioffe & Szegedy, 2015). As for the ViT-B/16 and ViT-L/16 algorithm, we incorporate R-Drop (Liang et al., 2021) to alleviate model distributional shift, a goal akin to that of MoReDrop. In all baselines, we set the dropout rate to 0.1, as recommended by the original papers, and this rate has been found to yield the best performance compared to other dropout rates in our evaluation, as shown in Appendix Fig. 3.

**Results.** The results displayed in Section 5.2 represent averages obtained from 5 independent seeds and experimental details are shown in Appendix B.3. When integrated with DropPath and DropBlock for the ResNet-18 model, MoReDrop consistently delivers superior performance on both the CIFAR-10 and the more challenging CIFAR-100 datasets. Notably, when paired with DropBlock, MoReDrop realizes a significant increase in accuracy compared to both the original ResNet-18 and DropBlock, with improvements of approximately $1\%$ in the CIFAR-10 dataset and $1.8\%$ in the more challenging CIFAR-100 dataset. The consistently improved performance across different dropout methods, i.e., DropPath and DropBlock, attests to the general applicability of MoReDrop.

With the backbone of ViT-B/16 and ViT-L/16, we find that MoReDrop outperforms the vanïlla model and its variant with R-Drop over three degrees of challenges tasks. The

*Table 3.* Accuracy on CIFAR-10, CIFAR-100 and ImageNet. Both MoReDrop and MoReDropL consistently outperform the baseline across all tasks. The best performances are in **bold**.

| Methods | CIFAR-10 | CIFAR-100 | ImageNet |
|---|---|---|---|
| ResNet-18 | $95.44_{\pm 0.07}$ | $77.78_{\pm 0.07}$ | - |
| + DropPath | $95.35_{\pm 0.05}$ | $78.12_{\pm 0.11}$ | - |
| + DropBlock | $95.53_{\pm 0.12}$ | $78.72_{\pm 0.06}$ | - |
| + MoReDropL | $95.79_{\pm 0.21}$ | $79.11_{\pm 0.05}$ | - |
| + DropPath + MoReDrop | $95.60_{\pm 0.14}$ | $79.25_{\pm 0.19}$ | - |
| + DropBlock + MoReDrop | $\mathbf{96.41}_{\pm 0.11}$ | $\mathbf{79.53}_{\pm 0.32}$ | - |
| ViT-B/16 | $98.68_{\pm 0.24}$ | $92.78_{\pm 0.10}$ | $84.05_{\pm 0.15}$ |
| + R-Drop | $98.97_{\pm 0.01}$ | $92.90_{\pm 0.02}$ | $84.16_{\pm 0.04}$ |
| + MoReDropL | $\mathbf{99.14}_{\pm 0.03}$ | $93.25_{\pm 0.03}$ | $84.62_{\pm 0.12}$ |
| + MoReDrop | $99.10_{\pm 0.06}$ | $\mathbf{93.38}_{\pm 0.04}$ | $84.43_{\pm 0.06}$ |
| ViT-L/16 | $99.15_{\pm 0.04}$ | $93.72_{\pm 0.03}$ | $84.68_{\pm 0.04}$ |
| + R-Drop | $99.11_{\pm 0.02}$ | $93.78_{\pm 0.05}$ | $84.85_{\pm 0.03}$ |
| + MoReDropL | $99.18_{\pm 0.07}$ | $93.74_{\pm 0.06}$ | $84.74_{\pm 0.06}$ |
| + MoReDrop | $\mathbf{99.24}_{\pm 0.02}$ | $\mathbf{93.91}_{\pm 0.01}$ | $84.91_{\pm 0.01}$ |

smaller margin gained ($< 1\%$) from MoReDrop and MoReDropL, compared with the backbone of ResNet-18, is attributed to the saturated performance by its ViT-B/16 and ViT-L/16 backbones. This observation underscores the superiority and scalability of our proposed algorithm on varying challenging tasks. Also, we observe that R-Drop necessitates approximately 3x the number of training epochs to converge on the ImageNet dataset, yet its final performance is not on par with our methods (both MoReDrop and MoReDropL). We attribute this to the compromise of model expressiveness for maintaining sub-model consistency in R-Drop with sub-to-sub regularization (detailed analysis in Section 5.4).

### 5.3. Image Generation Domains

**Benchmark Datasets.** Our evaluation extends to image generation tasks, utilizing the CIFAR-10 dataset for conditional image synthesis. The detailed introduction of diffusion models is in Appendix B.4. To measure the quality of generated images, we calculate the Fréchet Inception Distance (FID) (Heusel et al., 2017).

*Table 4.* Sample quality comparison on CIFAR-10.

| Method | FID |
|---|---|
| EDM (step=18) | $4.8786_{\pm 0.05}$ |
| + MoReDrop (step=18) | $\mathbf{4.3682}_{\pm 0.04}$ |
| EDM (step=30) | $4.7470_{\pm 0.01}$ |
| + MoReDrop (step=30) | $\mathbf{4.2274}_{\pm 0.03}$ |

**Model & Training.** In our comparative study, we utilize the state-of-the-art generative model EDM (Karras et al., 2022). Due to computational constraints, we preserved the original dropout ratio $p$ of 0.13 as specified in EDM, and set $\alpha = 1$ based on our empirical observations from other tasks, without further sweeping. Both EDM and EDM with MoReDrop were trained for 250 epochs.

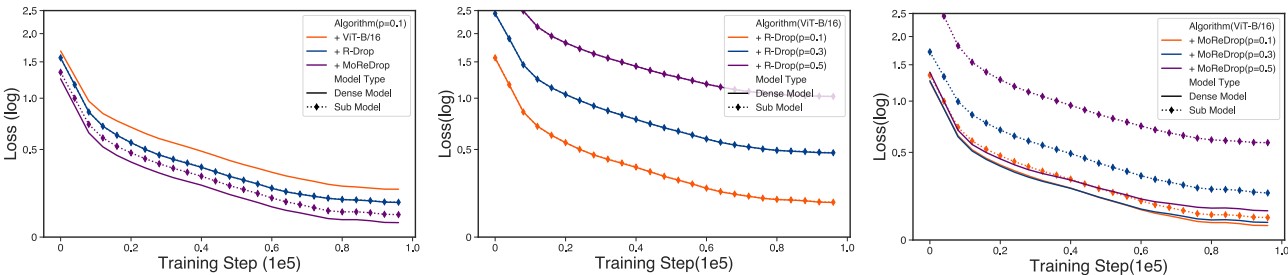

*Figure 2.* The training loss curves over ViT/16, comparing different dropout methods and rates on the `CIFAR-10` dataset. **Left:** Training loss curves of various methods with a consistent dropout rate ($p = 0.1$). **Middle:** Training loss curves of R-Drop under varying dropout rates. **Right:** Training loss curves of MoReDrop under different dropout rates.

**Results.** The experimental results, as presented in Table 4, using deterministic sampling with 18 and 30 steps, respectively, reveal that the application of MoReDrop consistently yields a reduction in FID. This pattern emphasizes the efficacy of MoReDrop in improving the quality and diversity of generated images. While systematic hyperparameter tuning is likely to refine and possibly enhance MoReDrop's performance on image generation tasks, we designate this avenue of exploration as future work due to computational budget constraints.

### 5.4. Dense-to-sub vs. Sub-to-sub Regularization

To further investigate the edges of the dense-to-sub regularization in MoReDrop, we perform a comprehensive comparative loss analysis between MoReDrop (dense-to-sub) and the state-of-the-art sub-to-sub model regularization approach, R-Drop. In Fig. 2, we present the training loss curves over ViT-B/16, comparing different rates on the `CIFAR-10` dataset. Note that we only track the MoReDrop sub-model R-Drop dense-model loss without gradient updates. Consistent with our expectations from Fig. 2 (Left), the training losses of dense models exhibit an inverse relationship with their performance as demonstrated in Section 5.2, where MoReDrop achieves the lowest training loss as well as the highest performance.

**zero-forcing discrepancy[2] of R-Drop.** The discrepancy of training curves between dense and sub-models in Fig. 2, under the same algorithm, approximates the model discrepancy in function space. While R-Drop presents negligible divergence between its dense and sub-models (blue lines), an observed divergence of those is shown in MoReDrop (purple lines). This finding indicates that a high level of consistency among sub-models may compromise model expressivity, leading to incomparable performance. Additionally, a key factor contributing to this inferior performance is the inherent expectation operator used in R-Drop during the evaluation process.

---

[2]The 'discrepancy' is specifically constrained to the training process. In contrast, 'model distributional shift' refers to differences arising from varying model configurations during training and evaluation phases.

**Does higher dropout rates mitigate zero-forcing discrepancy?** To answer it, we analyze the loss curves from dense and sub-models across varying dropout rates of R-Drop in Fig. 2 (Middle). We observe that the loss gap approaches zero irrespective of the dropout rate with $p \in [0.1, 0.3, 0.5]$. However, this is concurrent with increased loss and decreased performance, shown in Fig. 4, suggesting a strong performance compromise while constraining sub-model pairs. We hypothesize the loss of generalization in sub-to-sub regularization along R-Drop arises from the rising challenge to maintain consistency. In contrast, MoReDrop dense-to-sub regularization approach successfully preserves model generalization ability. This is evidenced by the observed positive correlation between the dropout rate $p$ and the loss gap, as illustrated in Fig. 2 (Right) and Fig. 4.

## 6. Conclusion

We present MoReDrop, an efficient method designed to counter model distributional shift in dropout models, capitalizing on dropout benefits, without dropping during gradient backpropagation. MoReDrop solely updates the dense model, maintaining uniformity across training and inference phases and focusing on its loss function optimization. MoReDrop then integrates a regularization loss originating from the output divergence from the dense model and sub-models. We highlight that the sub-models only passively update due to shared parameters to the dense model. For computational efficiency, MoReDropL, a variant of MoReDrop, limits the dropout and matrix multiplication to the final layer, balancing efficiency with generalization. Our experimental results across a variety of tasks and domains consistently demonstrate that both MoReDrop and MoReDropL achieve state-of-the-art performance in the majority of tasks and their strong robustness. Significantly, MoReDropL not only reduces computational demands but also outperforms MoReDrop on some tasks, such as the `RTE` task of the GLUE benchmark from language understanding domains. This finding paves the way for future research to delve into the joint effects of last-layer dropout and fine-tuning, while simple last-layer fine-tuning can match or outperform state-of-the-art approaches in image domains (Kirichenko et al., 2022; Le et al., 2023).

**Broader impact.** Our paper studies the model distributional shift problem in dropout models. By thinking a step further about the fundamental reason for the model distributional shift, our paper has the potential to encourage researchers to solve existing problems from a new perspective, distant from prior methods. This work also has potential impacts on the field of last-layer pre-training for giant models.

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

# A. Proof of Theorem 4.1

**Theorem A.1.** *The regularizer $\mathcal{R}$ (Eqn. (4)), under the condition $l(S; \theta) - l(\theta) \leq 0$, maintains non-negativity over the expectation over $S$ throughout the training process:*

$$\mathbb{E}_{S_i}\left[\frac{1}{N}\sum_{i=1}^{N}\frac{\exp(\alpha \cdot (l(S;\theta) - l(\theta))) - 1}{\exp(\alpha \cdot (l(S;\theta) - l(\theta))) + 1}\right] \geq 0$$

*Proof.* Given $f(x) = (e^{ax} - 1)/(e^{ax} + 1)$ is convex ($f''(ax) \geq 0$) with $a > 0$ when $x \leq 0$. We can apply Jensen's Inequality to Eqn. (4) under the condition $l(S; \theta) - l(\theta) \leq 0$:

$$\mathbb{E}_{S_i}\left[\frac{1}{N}\sum_{i=1}^{N}\frac{\exp(\alpha \cdot (l(S;\theta) - l(\theta))) - 1}{\exp(\alpha \cdot (l(S;\theta) - l(\theta))) + 1}\right] \geq \frac{\exp\left(\mathbb{E}_{S_i}\left[\frac{1}{N}\sum_{i=1}^{N}\alpha \cdot (l(S;\theta) - l(\theta))\right]\right) - 1}{\exp\left(\mathbb{E}_{S_i}\left[\frac{1}{N}\sum_{i=1}^{N}\alpha \cdot (l(S;\theta) - l(\theta))\right]\right) + 1}$$

We aim to establish the following inequality as the denominator is non-negative:

$$\exp\left(\mathbb{E}_{S_i}\left[\frac{1}{N}\sum_{i=1}^{N}\alpha \cdot (l(S;\theta) - l(\theta))\right]\right) - 1 \geq 0,$$

which simplifies to:

$$\mathbb{E}_{S_i}\left[\frac{1}{N}\sum_{i=1}^{N}(l(S;\theta) - l(\theta))\right] \geq 0, \tag{6}$$

where $\alpha$ is omitted as it holds positive. For a tractable approximation for dropout variable $p$, we use Bayes' rule to express the parameterized conditional probability of the output $y$ given the input $x$ and $p$ under parameter $\theta$:

$$p(y \mid x; \theta) = \int_{\mathcal{S}} p(y \mid x, s; \theta)p(s)d\mu(s).$$

Next, we reformulate the loss function for sub-models incorporating dropout (as in Eqn. 3):

$$\mathbb{E}_{S_i}\left[\frac{1}{N}\sum_{i=1}^{N}l(S;\theta)\right] = -\mathbb{E}_{S_i}\left[\frac{1}{N}\sum_{i=1}^{N}\log p(y_i|x_i, S_i; \theta)\right]$$

$$= -\int_{\mathcal{S}}\prod_{i=1}^{N}p(s_i)\left(\frac{1}{N}\sum_{i=1}^{N}\log p(y_i|x_i, s_i; \theta)\right)d\mu(s_1)\ldots d\mu(s_N)$$

$$= -\frac{1}{N}\sum_{i=1}^{N}\int_{\mathcal{S}}p(s_i)\log p(y_i|x_i, s_i; \theta)d\mu(s_i).$$

Given that $\log(\cdot)$ is a concave function, Jensen's Inequality yields:

$$\int_{\mathcal{S}}p(s)\log p(y|x, s; \theta)d\mu(s) \leq \log\int_{\mathcal{S}}p(s)p(y|x, s; \theta)d\mu(s).$$

Thus

$$\mathbb{E}_{S_i}\left[\frac{1}{N}\sum_{i=1}^{N}(l(S;\theta))\right] \geq -\frac{1}{N}\sum_{i=1}^{N}\log\int_{\mathcal{S}}p(s_i)p(y|x, s; \theta)d\mu(s_i) = \frac{1}{N}\sum_{i=1}^{N}l(\theta).$$

Consequently, the proof is finished as Eqn. (6) is satisfied with the condition $l(S; \theta) - l(\theta) \leq 0$.

$\square$

# B. Experiments Details

## B.1. Hardware Setup

Our experiments were performed using PyTorch and run on NVIDIA GeForce RTX 3090 and NVIDIA A800 80GB PCIe graphics cards. For the `CIFAR-10` and `CIFAR-100` tasks, we utilized single-card training on the NVIDIA GeForce RTX 3090. For the `GLUE` and `ImageNet` tasks, we employed distributed training across $4\times$ NVIDIA A800 80GB PCIe cards.

## B.2. Natural Language Processing

**Model Details.** Our experiments utilized three pre-trained models: BERT-base, RoBERTa-base and DeBERTaV3-xsmall.

BERT-base is a transformer-based model with 12 layers, 768 hidden units, and 12 attention heads, totaling 110 million parameters. It was pre-trained on a large corpus of English text from the BooksCorpus ($800M$ words) and English Wikipedia ($2,500M$ words).

RoBERTa-base, on the other hand, is a variant of BERT that uses a larger byte-level BPE vocabulary, longer training time, and different pre-training data. It has the same architecture as BERT-base but was trained on more data ($160GB$ of text).

DeBERTaV3-xsmall is an advancement in transformer-based models, incorporating disentangled attention and an enhanced mask decoder. With a more compact structure than its larger counterparts, it has 12 layers, 384 hidden units, and 12 attention heads, resulting in a total of around 22 million parameters. This model variant was pre-trained on a diverse set of textual data sources, similar to those used for BERT and RoBERTa, but with additional refinements in training techniques and optimizations that enhance its performance on downstream tasks, despite its smaller size.

**Datasets & Evaluation Metrics.** For language understanding tasks, we adhere to the prevalent pre-training and fine-tuning methodology, with the GLUE benchmark serving as the fine-tuning set. Consistent with previous studies, we focus on eight tasks, including single-sentence classification tasks (CoLA, SST-2), sentence-pair classification tasks (MNLI, QNLI, RTE, QQP, MRPC), and the sentence-pair regression task (STS-B). Detailed data statistics can be found in the original paper (Wang et al., 2019). The evaluation metrics for the aforementioned tasks are as follows: The STS-B task is evaluated using the Pearson correlation; The CoLA task is assessed via Matthew's correlation; Both the F-1 score and accuracy are used as metrics for the MRPC and QQP tasks; The remaining tasks (MNLI, QNLI, RTE, SST-2) are evaluated based on accuracy. These metrics collectively provide a holistic assessment of the models' performance across a range of language understanding tasks.

**Baseline Setting & Experiments Design.** Given that BERT-base, RoBERTa-base and DeBERTaV3-xsmall all incorporate standard dropout, MoReDrop can be directly applied. Meanwhile, MoReDropL was applied before the last linear layer as shown in Fig. 1.

**Fine-tuning Details.** All the models were fine-tuned on the GLUE benchmark datasets. As the datasets vary in size and complexity, we employed a dynamic dropout rate and used a set of different values for the parameter $\alpha$. The exact values of the dropout rate and $\alpha$ were determined based on the specific characteristics of each task. The chosen values for each task and model can be found in Table 11, Table 12 and Table 13.

**Hyperparameters & Training Setting.** For the fine-tuning process, we adhered to the original training hyperparameters used in the BERT, RoBERTa and DeBERTa models. For the sake of simplicity in implementation, we assigned the same values for batch size, learning rate, and epochs across different tasks and models, which were 32, $2e-5$, and 3, respectively. We used the Adam optimizer for both models. The precise hyperparameters for each task and model can be found in Table 11, Table 12 and Table 13.

## B.3. Image Classification

**Model Details.** Our image classification experiments employed three distinct models: ResNet-18, ViT-B/16 and ViT-L/16. ResNet-18, is a smalle model boasting 1.2 million parameters. As a member of the ResNet family, it utilizes a residual learning framework to streamline the training of networks, encompassing 18 layers that include convolutional, identity, and fully connected layers. ViT-B/16 and ViT-L/16, in contrast, are larger models with 86 million and 307 million parameters, respectively. These models are adaptations of the Vision Transformer (ViT), originally developed for natural language processing tasks, now repurposed for computer vision applications.

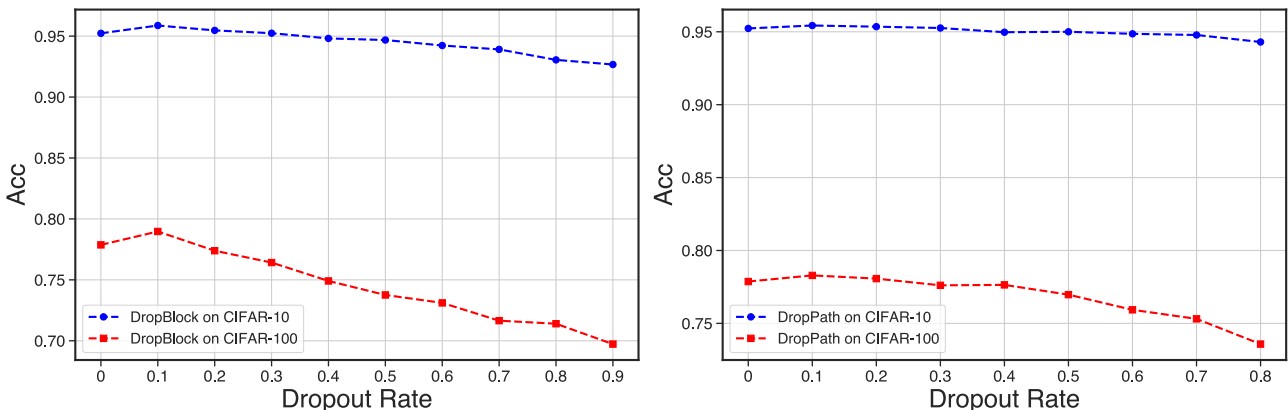

*Figure 3.* The accuracy of ResNet-18 employing DropBlock and DropPath under varying dropout rates on CIFAR-10 and CIFAR-100. **Left:** Accuracy curves of ResNet-18 utilizing DropBlock. **Right:** Accuracy curves of ResNet-18 utilizing DropPath. Note that DropPath failed to train the model under the dropout rate $p = 0.9$, hence this data point is not represented in the figure.

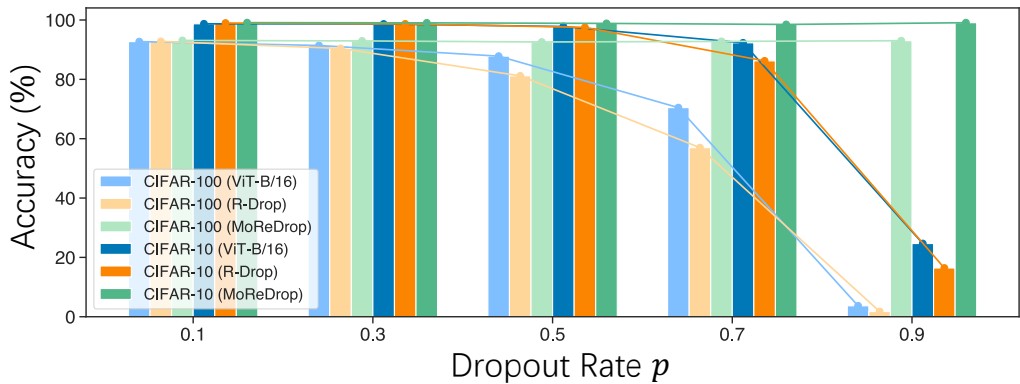

*Figure 4.* Accuracy across varying dropout ratios for different methods on CIFAR datasets.

**Baseline Setting & Experiments Design.** While the vanïlla ResNet-18 does involve the dropout technique, we take DropBlock (Ghiasi et al., 2018) and DropPath (Larsson et al., 2017) as our baselines. DropBlock mitigates overfitting by dropping continuous regions of neurons, while DropPath zeroes out an entire branch in the neural network during training, as opposed to just a single unit, making it a perfect match for ResNet. We set the dropout rate to 0.1, as recommended by the original papers for both DropBlock and DropPath, as well as the best dropout rate referring in our experiments(as shown in Fig. 3). This rate has been determined to yield the best performance in comparison to other dropout rates. Further details about these experiments can be found in Table 14.

In contrast, we adopt R-Drop as the comparison in ViT-B/16 and ViT-L/16, which minimizes the bidirectional KL-divergence of the output distributions of any pair of sub-models sampled from dropout in model training. As both ViT-B/16 and ViT-L/16 incorporate standard dropout in their architecture, MoReDrop is seamlessly applicable to these models.

For MoReDropL, as applied in Appendix B.2, we employed standard dropout before the final linear layer.

**Fine-tuning Details.** On ResNet-18, we fine-tuned the `CIFAR-10`, `CIFAR-100` while ViT-B/16 and ViT-L/16 on `CIFAR-10`, `CIFAR-100` and `ImageNet` datasets. As the datasets vary in size and complexity, we employed a dynamic dropout rate and used a set of different values for the parameter $\alpha$. The exact values of the dropout rate and $\alpha$ were determined based on the specific characteristics of each task. The chosen values for each task and model can be found in Table 14.

**Hyperparameters & Training Setting.** For the fine-tuning process, we used different training hyperparameters for each

model. For ResNet-18, we used a batch size of 128 for `CIFAR-10` and `CIFAR-100`. For ViT-B/16 and ViT-L/16, we used a batch size of 32 for `CIFAR-10`, and 256 for `CIFAR-100` and 64 for `ImageNet`. Regarding the training epochs, ResNet-18 was set to 200 for both `CIFAR-10` and `CIFAR-100`. For ViT-B/16, we set the training epochs to 50 for `CIFAR-10` and `CIFAR-100`, and 10 for `ImageNet` in the original ViT-B/16 and our algorithm. Due to the difficulty in achieving convergence with R-Drop, we increased the training epochs to 30 when training `ImageNet`. Furthermore, the image size for ViT-B/16 was set to 384 for `ImageNet`, and 224 for both `CIFAR-10` and `CIFAR-100`.

The exact hyperparameters for each task and model can be found in Table 14. For the `CIFAR-10` and `CIFAR-100` tasks, we employed the SGD optimizer with a learning rate of $1e - 2$, while for the `ImageNet` task in ViT-B/16, we utilized the Adam optimizer with a learning rate of $1e - 4$ and $5e - 5$ for ViT-L/16. For data augmentation, we exclusively utilized random cropping for the `CIFAR-10` and `CIFAR-100` tasks. However, for the `ImageNet` dataset, we adopted RandAugment (Cubuk et al., 2020).

## B.4. Image Generation Domains

**Introduction.** Diffusion models (Sohl-Dickstein et al., 2015; Ho et al., 2020; Song et al., 2020) are a novel class of generative models that produce high-quality samples, excelling in tasks such as image and audio generation. Inspired by the natural diffusion process, these models convert data into noise through a series of steps and then learn to reverse this process. By predicting and cancelling out the noise incrementally, the models reconstruct the original data, generating new samples. While diffusion models are powerful in capturing complex data patterns and ensuring the realism of the samples, they often require significant computational power due to the numerous steps.

**Experimental Setup.** We employ one of the state-of-the-art generative models, EDM (Karras et al., 2022), as our training backbone. Since EDM uses standard dropout, MoReDrop can be readily applied. Due to computational resource constraints, we maintained the original dropout ratio $p = 0.13$ and assigned $\alpha = 1$.

## B.5. Training Time

In Table 5, we present the per epoch training time for MoReDrop, MoReDropL, and baselines on the `CIFAR-10` task, executed on a GPU (NVIDIA A800 80GB PCIe) using a batch size of 32. MoRe-DropL exhibits a significant efficiency advantage with nearly the same training time as the backbone, and approximately $50\%$ faster than both MoReDrop and R-Drop. For example, in the case of ViT-B/16, R-Drop and MoReDrop require 176s and 172s respectively to execute a training epoch, whereas MoReDropL requires only 92s, almost on par with the $90s$ needed for ViT-B/16.

*Table 5.* Training time per epoch.

| Model | Training time (seconds) |
|---|---|
| ResNet-18 | 14 |
| + MoReDropL | 15 |
| + MoReDrop | 29 |
| ViT-B/16 | 90 |
| + R-Drop | 176 |
| + MoReDropL | 92 |
| + MoReDrop | 172 |

## B.6. Ablation Study on Hyperparameters

In this section, we conduct an ablation study to evaluate the impact of hyperparameters in MoReDrop, specifically the dropout ratio $p$ and the regularization weight $\alpha$. The sets explored for $p$ and $\alpha$ are $\{0.1, 0.3, 0.5, 0.7\}$ and $\{0.1, 0.5, 1, 2\}$, respectively. As illustrated in Fig. 5, MoReDrop exhibits significant robustness to hyperparameter variations. Most combinations yield performance superior to the baseline, with the exception of certain extreme combinations such as $(p, \alpha) = (0.7, 2)$ on the `CIFAR-10` dataset using the ResNet-18 backbone, and $(p, \alpha) = (0.7, 1)/(0.5, 2)/(0.5, 1)$ on the `MRPC` task. Generally, when tuning MoReDrop for application in other domains, we recommend prioritizing adjustments to $p$ due to its significant influence on performance.

Typically, higher $p$ values, e.g., 0.5, are associated with less precise predictions, shown in Fig. 4. However, MoReDrop challenges this norm by exhibiting superior performance even at high dropout, such as $p = 0.7$. This notable achievement can be ascribed to two main contributing factors.

Firstly, the performance in MoReDrop primarily relies on traditional dense training for gradient updates, upon which further enhancements are expected through this regularization. Secondly, the inherent bounded nature of $\mathcal{R}$, along with the weight factor $\alpha$, ensures that its influence on the primary optimization target remains controlled. To counter underperforming sub-models with high dropout rates $p$, lowering $\alpha$ is crucial for diminishing the influence of $\mathcal{R}$, thus preserving the performance anchored in the dense model. Conversely, a higher coefficient $\alpha$ is beneficial at a lower dropout ratio $p$, capitalizing on

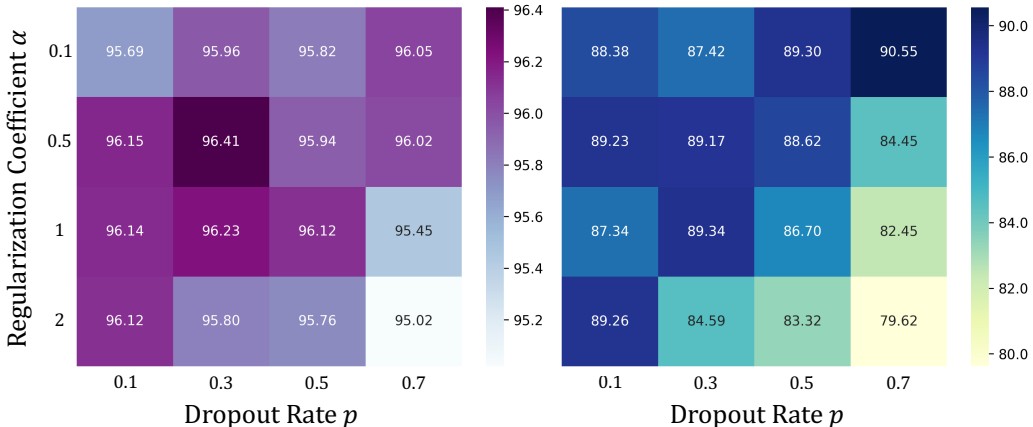

*Figure 5.* Performance of various tasks using MoReDrop under different hyperparameter combinations. **Left:** Performance of ResNet-18 with DropBlock and MoReDrop on the `CIFAR-10` dataset. **Right:** Performance of RoBERTa-base with MoReDrop on the `MRPC` task.

sub-model strengths from dropout, as demonstrated in Fig. 5. We highlight that the ablation study targets high $p$ to showcase MoReDrop's robustness; however, lower $p$ values are generally favored in practice.

### B.7. Ablation Study on Different Regularization Forms

In prior research, specifically in the contexts of ELD and FD, the $L_2$ distance on hidden states was employed as a regularization loss function. However, this approach diverges significantly from the primary training objective, which is to minimize the negative log-likelihood over the model's output distribution. R-Drop introduced the use of KL-divergence between output probability distributions. While this method imposes a strong constraint on the model, affecting its generalization capabilities, it also incurs considerable computational costs.

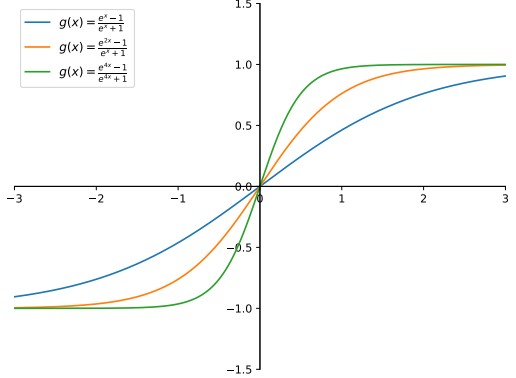

*Figure 6.* Illustration of the $g(x)$.

To address these challenges, we propose to use a loss scalar to quantify the discrepancy and then project this into a relative space, adjustable through a hyperparameter $\alpha$. Additionally, we exploit the bounded nature of the $sigmoid(\cdot)$ function, within $[0, 1]$, to reshape it into $g(x) = \frac{e^x - 1}{e^x + 1} = 2 \cdot (sigmoid(x) - \frac{1}{2})$, forming our regularization loss function. Fig. 6 illustrates the behavior of $g(x)$ for various values of $\alpha$, highlighting how the function's shape is modulated to control the impact on the regularization term.

We conduct numerical experiments to demonstrate the efficacy of different formulations of $g(\cdot)$ For practicality and simplicity in implementation, we adopt ResNet-18 as our model backbone. We configure four distinct experimental scenarios: ResNet-18-2D, which incorporates 2 dropout layers.

*Table 6.* Results of different form of $g(\cdot)$ on CIFAR-10 with ResNet-18.

| $g(\cdot)$ | $p = 0.1$ | $p = 0.3$ | $p = 0.5$ | $p = 0.7$ | $p = 0.9$ |
|---|---|---|---|---|---|
| $g(x) = x$ | 95.22 | 95.31 | 95.13 | 95.04 | 90.07 |
| $g(x) = x^2$ | 95.42 | 95.61 | 95.33 | 95.23 | 94.23 |
| $g(x) = \frac{e^x - 1}{e^x + 1}$ | **95.56** | **95.70** | **95.72** | **95.47** | **95.23** |

The tables above reveal distinct performance trends across different $g(\cdot)$ functions. Specifically, $g(x) = \frac{e^x - 1}{e^x + 1}$ outperforms

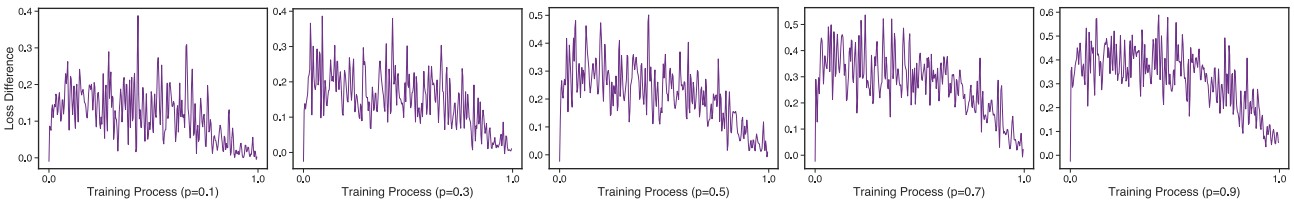

*Figure 7.* Comparison of the expected loss discrepancies between sub-models and the dense model across different dropout ratios for ResNet-18 on CIFAR-10 during training. The loss gap remains positive after the initial phase, attributable to the randomness in model weight initialization.

in all settings, $g(x) = x$ demonstrated inferior performance compared to both $g(x) = x^2$ and $g(x) = \frac{e^x - 1}{e^x + 1}$. Notably, $g(x) = x^2$ and $g(x) = \frac{e^x - 1}{e^x + 1}$ demonstrate comparable effectiveness under conditions of low dropout rates. However, as dropout rates increase to $0.7, 0.9$, the function $g(x) = \frac{e^x - 1}{e^x + 1}$ exhibits superior robustness in these extreme conditions. This leads to the conclusion that $g(x) = \frac{e^x - 1}{e^x + 1}$ is not only adaptable to varying scenarios but also demonstrates enhanced performance and robustness across different dropout rates.

Further investigation focuses on the role of the hyperparameter $\alpha$ within our regularization function, examining whether it is more effective in the form of $\alpha \cdot g(x)$ or $g(\alpha \cdot x)$. To this end, an experiment was conducted using ResNet-18-4D as the backbone, setting $p = 0.5$, and observing performance variations across different values of $\alpha$.

*Table 7.* Results under different $\alpha$ and $g(\cdot)$ on CIFAR-10 with ResNet-18.

| $g(\cdot)$ | $\alpha = 0.1$ | $\alpha = 0.5$ | $\alpha = 1$ | $\alpha = 2$ | $\alpha = 5$ | $\alpha = 10$ |
|---|---|---|---|---|---|---|
| $\alpha \cdot g(x)$ | **95.55** | 95.81 | 95.61 | **95.48** | 94.28 | 73.80 |
| $g(\alpha \cdot x)$ | 95.52 | **95.83** | **95.66** | **95.48** | **95.36** | **95.22** |

From the table, we observe that both $\alpha \cdot g(x)$ and $g(\alpha \cdot x)$ deliver comparable performance under low $\alpha$ value. However, with the increment of $\alpha$, $g(\alpha \cdot x)$ performs more robust compared to $g(\alpha \cdot x)$, meanwhile considering various application scenarios, it becomes evident that a bounded loss function like $g(\alpha \cdot x)$ offers more versatility in addressing a range of issues. Therefore, this paper opts for the use of $g(\alpha \cdot x)$ in the proposed framework. Nevertheless, it is crucial to recognize the potential for discovering more effective loss functions. Future research will be directed towards exploring these alternatives to further enhance the performance and adaptability of our model.

### B.8. Baseline Design

The absence of certain experimental results was a deliberate design choice. We provide a detailed explanation as follows:

**Why R-Drop is our main baseline for comparison?** Our decision to use R-Drop as the main baseline is influenced by its proven computational efficiency and benchmarking superiority over ELD (Ma et al., 2017) and FD (Zolna et al., 2018), as shown in the R-Drop paper. This choice follows the R-Drop paper, which compares R-Drop and FD based on FD's established advantages over ELD, thereby focusing our comparisons on R-Drop versus MoReDrop.

Regarding WordReg (Xia et al., 2023), the absence of publicly available code and detailed experimental information hinders direct comparison. We have reached out to the authors of WordReg via email to inquire further without a response. On the other hand, WordReg and R-Drop share a similar paradigm in using regularization to mitigate model distributional shifts. Conversely, MoReDrop represents a novel paradigm, aiming to harness the advantages of dropout through regularization while the models distributional shift is 0 through actively updating the dense model.

### B.8.1. NLP DOMAINS IN TABLE 2

**Why is R-Drop excluded?** We observed that R-Drop requires significant time (3x-10x) to converge, which is consistent with the experimental details shown in Table 8 of the R-Drop paper. To make a fair comparison to the established baseline,

we meticulously adhered to the experimental setup outlined in the BERT (Devlin et al., 2019) paper, encompassing 3 identical training epochs. In Table 8, we provide the best performance of R-Drop by sweeping the training epochs from 3 to 30. We observe that R-Drop fails to converge within the 3 epochs, which is consistent with Table 10 in the R-Drop paper. For example, the RTE dataset requires 30 epochs to achieve results comparable to ours, and other datasets need 10 epochs or more to converge. Therefore, considering a fair comparison in the main part, we exclude R-Drop for the comparison. Given the computational cost, we list only the relatively small datasets CoLA, MRPC, STS-B, and RTE in Table 8. For instance, training on the QQP dataset requires nearly 40 hours using 4 A100 80GB GPUs with R-Drop applied.

**Why does R-Drop need more training time to coverage?**    Our experiments reveal that, despite a substantial increase in computational resources, performance significantly lags behind expectations. We attribute this leading reason to the compromise of model expressiveness for maintaining sub-model consistency (detailed analysis in Section 5.4).

*Table 8.* Performance comparison across different models and configurations in NLP tasks.

| Method | CoLA | MRPC | STS-B | RTE |
|---|---|---|---|---|
| BERT-base | 56.49 | 85.10/89.41 | 87.92 | 67.99 |
| R-Drop | 56.70 | 86.46/90.39 | 87.98 | 68.31 |
| MoReDrop | **58.99** | **87.18/90.86** | **88.31** | **69.98** |
| Roberta-base | 60.07 | 87.50/90.84 | 89.68 | 72.77 |
| +R-Drop | 61.75 | 88.86/92.02 | **90.66** | 76.36 |
| +MoReDrop | **62.37** | **89.80/92.44** | 90.55 | **77.45** |
| DeBERTaV3-xsmall | 61.34 | 89.58/92.29 | 89.95 | 71.02 |
| +R-Drop | 62.78 | 89.90/92.20 | **90.34** | **76.52** |
| +MoReDrop | **64.64** | **90.04/92.74** | 90.32 | 76.46 |

### B.8.2. IMAGE DOMAINS IN SECTION 5.2

**Why ResNet-18 with MoReDrop and ResNet-18 with R-Drop are excluded?**    The integration of dropout with BatchNorm in the standard ResNet architecture has been shown to degrade performance, as reported in (He et al., 2016; Zhou et al., 2020). Our findings corroborate this, as detailed in Table 9. To ensure a fair comparison, we maintain the original architecture of ResNet18 unchanged. Our experiments demonstrate that incorporating dropout into ResNet-18 adversely affects its performance. Introducing dropout alongside R-Drop in ResNet18 for `CIFAR` tasks leads to catastrophic failure, whereas MoReDrop remains robust due to its primary dense model optimization.

*Table 9.* Performance of R-Drop and MoReDrop with ResNet18 on CIFAR tasks.

| Method | CIFAR-10 | CIFAR-100 |
|---|---|---|
| ResNet18 | 95.44 | 77.87 |
| + dropout | 95.32 | 77.69 |
| + dropout + R-Drop ($\alpha = 0.5$) | 12.02 | 1.86 |
| + dropout + R-Drop ($\alpha = 0.1$) | 11.50 | 1.70 |
| + dropout + R-Drop ($\alpha = 0.01$) | 14.28 | 1.85 |
| + dropout + MoReDrop ($\alpha = 0.5$) | **95.51** | **78.20** |

**Why ResNet-18 + DropPath + R-Drop and ResNet-18 + DropBlock + R-Drop are excluded?**    In our experiments, We observe that R-Drop fails to achieve convergence and even is not comparable to the vanilla backbone models. We hypothesize this phenomenon for two main interplayed reasons: (a) DropBlock and DropPath employ a form of dropping that is fundamentally different from traditional dropout, leading to a significant feature loss even at low dropout rates (e.g., p=0.1), when combined with DropBlock. (b) Maintaining sub-model consistency potentially limits model expressivity, resulting in suboptimal performance. Additionally, the complexity of maintaining sub-model consistency increases with the varying dimensions of feature dropping.

On the other hand, MoReDrop can guarantee coverage as the parameter optimization anchors the dense model. Our comprehensive analysis contrasts MoReDrop's dense-to-sub approach with R-Drop's sub-to-sub methodology, detailed in Section 5.4, underscores our method's superior scalability. We also present a new experiment under ViT-B + dropout head (p=0.1), a dropout variant by dropping attention heads in a multi-head attention mechanism [3]. It is similar to DropPath and DropBlock, dropping some structural features of the model. As shown in Table 10, adding the R-Drop leads to catastrophic failure even not comparable to the backbone.

*Table 10.* Performance of R-Drop and MoReDrop with ViT-B/16 on CIFAR tasks.

| Method | CIFAR-10 | CIFAR-100 |
|---|---|---|
| ViT-B/16 | 98.68 | 92.78 |
| + DropHead | 99.11 | 93.25 |
| + R-Drop + DropHead | 99.00 | 91.96 |
| + MoReDrop + DropHead | **99.14** | **93.51** |

*Table 11.* Hyperparameters for BERT-base experiments.

| Datasets | $p$ | | | $\alpha$ | | | Learning Rate | Batch Size | Epochs |
|---|---|---|---|---|---|---|---|---|---|
| | BERT-base | + MoReDropL | + MoReDrop | BERT-base | + MoReDropL | + MoReDrop | | | |
| CoLA | 0.1 | 0.7 | 0.2 | - | 1 | 1 | 2e-5 | 32 | 3 |
| SST-2 | 0.1 | 0.3 | 0.2 | - | 1 | 1 | 2e-5 | 32 | 3 |
| MRPC | 0.1 | 0.9 | 0.3 | - | 2 | 0.1 | 2e-5 | 32 | 3 |
| STS-B | 0.1 | 0.3 | 0.1 | - | 0.1 | 1 | 2e-5 | 32 | 3 |
| QQP | 0.1 | 0.2 | 0.2 | - | 0.5 | 0.5 | 2e-5 | 32 | 3 |
| MNLI | 0.1 | 0.2 | 0.2 | - | 1 | 1 | 2e-5 | 32 | 3 |
| QNLI | 0.1 | 0.3 | 0.3 | - | 1 | 1 | 2e-5 | 32 | 3 |
| RTE | 0.1 | 0.5 | 0.3 | - | 0.1 | 0.5 | 2e-5 | 32 | 3 |

*Table 12.* Hyperparameters for RoBERTa-base experiments.

| Datasets | $p$ | | | $\alpha$ | | | Learning Rate | Batch Size | Epochs |
|---|---|---|---|---|---|---|---|---|---|
| | RoBERTa-base | + MoReDropL | + MoReDrop | RoBERTa-base | + MoReDropL | + MoReDrop | | | |
| CoLA | 0.1 | 0.5 | 0.2 | - | 0.1 | 1 | 2e-5 | 32 | 3 |
| SST-2 | 0.1 | 0.3 | 0.2 | - | 0.1 | 1 | 2e-5 | 32 | 3 |
| MRPC | 0.1 | 0.1 | 0.7 | - | 2 | 0.1 | 2e-5 | 32 | 3 |
| STS-B | 0.1 | 0.2 | 0.4 | - | 2 | 2 | 2e-5 | 32 | 3 |
| QQP | 0.1 | 0.3 | 0.2 | - | 0.5 | 0.5 | 2e-5 | 32 | 3 |
| MNLI | 0.1 | 0.3 | 0.2 | - | 0.5 | 1 | 2e-5 | 32 | 3 |
| QNLI | 0.1 | 0.1 | 0.2 | - | 0.1 | 1 | 2e-5 | 32 | 3 |
| RTE | 0.1 | 0.1 | 0.2 | - | 0.1 | 0.1 | 2e-5 | 32 | 3 |

*Table 13.* Hyperparameters for DeBERTaV3-xsmall experiments.

| Datasets | p DeBERTaV3-xsmall | + MoReDropL | + MoReDrop | α DeBERTaV3-xsmall | + MoReDropL | + MoReDrop | Learning Rate | Batch Size | Epochs |
|---|---|---|---|---|---|---|---|---|---|
| CoLA | 0.1 | 0.1 | 0.1 | - | 0.1 | 0.5 | 2e-5 | 32 | 3 |
| SST-2 | 0.1 | 0.3 | 0.3 | - | 1 | 0.5 | 2e-5 | 32 | 3 |
| MRPC | 0.1 | 0.1 | 0.3 | - | 0.1 | 0.5 | 2e-5 | 32 | 3 |
| STS-B | 0.1 | 0.3 | 0.1 | - | 1 | 0.5 | 2e-5 | 32 | 3 |
| QQP | 0.1 | 0.5 | 0.1 | - | 1 | 0.5 | 2e-5 | 32 | 3 |
| MNLI | 0.1 | 0.3 | 0.2 | - | 2 | 1 | 2e-5 | 32 | 3 |
| QNLI | 0.1 | 0.3 | 0.1 | - | 2 | 2 | 2e-5 | 32 | 3 |
| RTE | 0.1 | 0.1 | 0.1 | - | 0.5 | 0.1 | 2e-5 | 32 | 3 |

*Table 14.* Hyperparameters for Image Classification tasks.

| Model | p CIFAR-10 | CIFAR-100 | ImageNet | α CIFAR-10 | CIFAR-100 | ImageNet | Batch Size CIFAR-10 | CIFAR-100 | ImageNet | Epochs CIFAR-10 | CIFAR-100 | ImageNet |
|---|---|---|---|---|---|---|---|---|---|---|---|---|
| ResNet-18 | - | - | - | - | - | - | 128 | 128 | - | 200 | 200 | - |
| + DropPath | 0.1 | 0.1 | - | - | - | - | 128 | 128 | - | 200 | 200 | - |
| + DropBlock | 0.1 | 0.1 | - | - | - | - | 128 | 128 | - | 200 | 200 | - |
| + MoReDropL | 0.1 | 0.3 | - | 0.1 | 0.5 | - | 128 | 128 | - | 200 | 200 | - |
| + DropPath + MoReDrop | 0.5 | 0.1 | - | 0.1 | 0.1 | - | 128 | 128 | - | 200 | 200 | - |
| + DropBlock + MoReDrop | 0.3 | 0.2 | - | 0.5 | 0.5 | - | 128 | 128 | - | 200 | 200 | - |
| ViT-B/16 | 0.1 | 0.1 | 0.1 | - | - | - | 32 | 256 | 64 | 50 | 50 | 10 |
| + R-Dorp | 0.1 | 0.1 | 0.1 | 0.3 | 0.6 | 0.6 | 32 | 256 | 64 | 50 | 50 | 30 |
| + MoReDropL | 0.1 | 0.4 | 0.1 | 1 | 1 | 1 | 32 | 256 | 64 | 50 | 50 | 10 |
| + MoReDrop | 0.1 | 0.1 | 0.1 | 1 | 0.5 | 1 | 32 | 256 | 64 | 50 | 50 | 10 |
| ViT-L/16 | 0.1 | 0.1 | 0.1 | - | - | - | 32 | 256 | 24 | 50 | 50 | 5 |
| + R-Dorp | 0.1 | 0.1 | 0.1 | 0.3 | 0.6 | 0.6 | 32 | 256 | 24 | 50 | 50 | 5 |
| + MoReDropL | 0.1 | 0.4 | 0.1 | 1 | 1 | 1 | 32 | 256 | 24 | 50 | 50 | 5 |
| + MoReDrop | 0.1 | 0.1 | 0.2 | 1 | 0.5 | 1 | 32 | 256 | 24 | 50 | 50 | 5 |

