# OpenReview forum: "MoReDrop: Dropout without Dropping"
_ICML.cc/2024/Workshop/WANT — WANT@ICML 2024 Poster_

### Official Review · Reviewer_e1EW · 2024-06-07
**Review of "MoReDrop: Dropout without Dropping"**

**Confidence:** 4

**Summary:**

The paper proposes a regularisation method for machine learning methods that forces the model weights to be similar to those obtained when using dropout. It is thus a regularisation alternative to dropout.

I like the idea, and the results look good, but I am missing some details, some clarity, and some comparisons.

**Strengths:**

I think the method is well-presented and the idea clear. The results appear favourable for the proposed method.

**Weaknesses:**

There are quite some things that could be made more clear in the paper, or explained better. For instance:

 - Already in the abstract, you start to talk about "the dense model", without explaining/defining what this means in this context.
 - You talk about distribution shift already in the abstract without explaining/defining what this means in this context. This should be explained early on, because you return to this in the introduction without explanation. Perhaps even illustrate the distribution shift, so that the problem becomes clear.
 - The abstract talk about divergence between the dense and dropout models, but (KL) divergence is what R-Drop used, you specifically propose something else.
 - You say that the other methods fail to prevent distribution shift because they apply "the expectation operator during evaluation", but no explanation/proof/illustration is presented. Of course in practice these other methods implement the expectation using stochastic samples, but in the limit the results should be the same (I would assume).
 - Around line 40, right column: You say that distances are computed in the other method, but never (until later) say what those distances are between.
 - What do you mean on line 89 when you say the regularised "functions as a toolkit"?

Major comments:
 - Line 108, right: The "dropout brings inconsistency" is note clear to me.
 - Line 135, left: Not clear what the significant difference would be.
 - Line 127, right: Should be 0 to L, right?
 - All: l is both a layer index and a loss function. Also, l is defined in two ways as a loss function. Be consistent and avoid ambiguities.
 - Line 142, right: Not setting weights, but setting activations to zero.
 - Line 148, right: The h is first a vector, and then as a function.
 - Line 155, right: The transform function has an index, but the weights in it does not.
 - Equation 1: You say that the expected H is the output of the ensemble, but previous work has concluded that the ensemble approximates a geometric mean over the subnetworks (see e.g., https://papers.nips.cc/paper_files/paper/2013/hash/71f6278d140af599e06ad9bf1ba03cb0-Abstract.html, https://arxiv.org/abs/1312.6197, or https://www.ncbi.nlm.nih.gov/pmc/articles/PMC3996711/).
 - Equation 1: You say that the expected dropout mask is the one used for the "dense network", but that would be a "mask" with all elements being 1-p. The input there should instead be a mask with all ones, since no elements are removed.
 - Don't use colon (:) before equations. Make the equations part of the sentences.
 - Equations 2 and 3: The left hand side doesn't use the index i.
 - The scaled version of the logistic sigmoid looks like just the tanh? But not quite, since it seems like 1 minus the numerator. Something's off there. Please double-check that you got it right.
 - I assume the regularisation term, R, is a function of \theta? This should be made clear in both Equations 4 and 5.
 - Line 242: There's not x and y in the statement.
 - It seems strange to me that you rely in the expectation of the regulariser to be positive, and then minimise that. That will probably work well, and seems to work well, but why not formulate a regulariser that immediately does what you actually want it to do? Instead of allowing, as you say, undesirable solutions, in the first place? The squared distance seems more reasonable, but that's fraternal dropout then. Please motivate your choice clearly. It does not look at a first glance that it matches any criterion, but there's a connection between the sigmoid of the expected ensemble output and the geometric mean. Perhaps it is that you actually makes the geometric means agree? See the links above.
 - The experiments seem non-exhaustive to me. I would like to see a clear comparison to at least standard dropout, R-Drop, and your method for all datasets.
 - Explain somewhere what the different font mean in the tables.
 - If I understand correctly (please explain this clearly), the standard errors come from running the same experiments multiple times with different initial weights? Note that this does not capture any aleatoric uncertainty stemming from the data, which is usually what you would want to present in maximum likelihood learning. You probably want to do some kind of resampling, or capture the uncertainty in the mean scores that you present.
 - Table 4: Why four significant digits here? Do remove two so that you have two here as well.
 - It will not be a fair comparison if you do a grid search for your method and use literature values for the competing methods. You need to do the same type of grid searches for all methods.
 - It seems that in most cases, the proposed method is added on top of R-Drop, or did I misunderstand this? If so, you cannot say that it is your method that is better. If so, do run the baseline method without dropout and without R-Drop, and only with your method, and then compare to using only dropout and using only R-Drop.
 - How were other hyper-parameters selected, such as the number of epochs?
 - Line 446, left: Do you mean re-training/fine-tuning?
 - Line 544: Explain the step where you removed the product. Also, note that you have two index i. Use another index for one of them.
 - Do add uncertainties (standard errors) also to the table sin the appendix.
 - Note that most of your results in the tables actually doesn't show significant differences between the different methods. You can only claim that your method actually works better in the cases when you have a significant difference.

Minor comments:
 - Line 18, right column: Dropout always uses Bernoulli distributed masks, not in general. Other versions exist, such as Gaussian Dropout, but that's another method with other properties.
 - Line 27, right column: What do you mean by assembling sub-models?
 - Page 1, footnote: I don't understand it. Please clarify or remove if not critical.
 - Line 96, right: Sentence messed up.
 - Line 127, left: "the on".
 - Line 130, left: "firstly find".
 - Line 123, right: "respective ... respectively".
 - All: M_i is all in bold, so the index i is also bold.
 - After Equation 1: Saying RHS and LHS of the minus is a but contrived. Perhaps just say the first and second terms on the left hand side?
 - Line 190, right: Comma at the end of the equation, should be full stop.
 - Line 432, right: No space before RTE.

**Limitations:**

The experiments need to be clarified and probably extended. Also, the motivation for the regulariser is not clear. See my more extensive comments under weaknesses.

**Suggestions:**

See my comments under weaknesses.

---

### Official Review · Reviewer_LamD · 2024-06-10

**Confidence:** 4

**Summary:**

The paper proposes an alternative (MoReDrop ) to dropout, as a way to mitigate the distributional shift between training and inference. In their method, the loss is calculated with and without dropout (model loss and sub-model loss), and the weight update is performed in relation to both. A lighter approach (MoReDropL), that only performs single forward-propagation, is also introduced.

The experimental results presented in the paper show improvement for a wide range of tasks, including text (BERT family), image classification (ViT-B/16 and Resnet-18, CIFAR / Imagenet), and CIFAR-10 image generation.

**Strengths:**

* Targets dropout, an important cornerstone of the model-training
* Good Empirical results, on a large variety of tasks
* Feasible, easy-to-implement method. Computational costs are discussed and acknowledged.

**Weaknesses:**

* I found Section 4 (the main section describing the method) hard to follow and would suggest rewriting 4.1 and 4.2. The description of MoReDrop is unnecessarily messy: $\mathcal{R}$ is suddenly introduced without explaining that it is a regularization term and the discussion of KL/L2 divergence is poorly connected to the rest of the section. There are more cases like this, which makes it harder to understand the method (e.g. the line starting with "Unlike conventional methods")

* The main motivation for using the new method is the distributional shift of dropout, and the paper does a decent job describing it. However, I didn't see an argument for why distributional shift is a problem. To me, it makes sense that regularization methods (Dropout, as well as data augmentation or mixup) would cause a distributional shift: after all, a common interpretation of these methods is that they try to make training "harder". It is common for models to perform worse during training for the same reason.
I also remember previous works about dropout [1] claiming that using an ensemble of dropout submodels results in worse results than what you get with standard dropout, despite having seemingly lower distributional shift.

* In the case of BeRT, there was a large HP scan, but the details are not entirely clear to me, so I am not sure whether the comparison is fair. Other experiments seem to have more robust hyper-parameters, and an ablation study is included in the supplementary.

[1]  The Implicit and Explicit Regularization Effects of Dropout:  https://arxiv.org/abs/2002.12915

**Suggestions:**

1. Rewrite 4.1 and 4.2
2. In Algorithm 1, you first run the dense model, followed by the sub-model. This will result in higher peak memory since when running the sub-model, you must also keep the activations of the dense model in memory (you will perform back-propagation later). This memory can be saved by simply running the sub-model first-- it runs on ``detach'' mode and we don't need to keep anything except the loss in the end.
3. Any changes that would strengthen the connection between the theoretical reasoning (distributional shift) and the empirical result would be helpful.

---

### Official Review · Reviewer_2BPN · 2024-06-11

**Confidence:** 3

**Summary:**

This study investigates the problem of model distributional shift between training and evaluation stages when using Dropout, and proposes a new approach, MoReDrop, which solely updates the dense model parameters and targets for model consistency applied throughout both the training and inference stages. The authors test the effectiveness of MoReDrop on various models and tasks.

Pros:

- The paper supports its contributions with extensive experimental evaluations across many benchmarks.
- The idea is simple and easy to understand
- The idea is novel

Cons:

- The performance improvement is marginal while it takes longer training time.
- Lack more baselines to valid the advantage of the propose method.

**Strengths:**

- idea is simple and straightforward
- extensive evaluation on multiple settings

**Weaknesses:**

- lack important baselines

---

### Meta-Review · Area_Chair_7Lsb · 2024-06-18

**Recommendation:** Accept (Oral)
**Confidence:** 4

**Metareview:**

**Strengths**
- Updating the dense model and incorporating dropout regularization through loss function are interesting differentiations from prior work.
- The proposed idea is intuitive, easy to implement, and promising.
- The evaluation is fairly extensive and includes a decent explanation of the results.

**Weaknesses**
- Evaluation is missing important baselines
- The writing could be improved in multiple places, as detailed by the reviewers.

**Summary***
I think novelty of the idea and the decent evaluation makes this an interesting work for the community.

---

### Decision · Program_Chairs · 2024-06-18

**Decision:**

Accept (Poster)

**Comment:**

We thank the authors for their time and contribution to WANT and we are pleased to share that after the reviewing process the paper has been accepted. Congratulations! We encourage the authors to consider reviewers' feedback for the improvement of the camera-ready version. We hope to see you in person at the workshop and brainstorm on efficient training research together!